# POLYNOMIAL-BASED SELF-ATTENTION FOR TABLE REPRESENTATION LEARNING

## ABSTRACT

Structured data, which constitutes a significant portion of existing data types, has been a long-standing research topic in the field of machine learning. Various representation learning methods for tabular data have been proposed, ranging from encoder-decoder structures to Transformers. Among these, Transformer-based methods have achieved state-of-the-art performance not only in tabular data but also in various other fields, including computer vision and natural language processing. However, recent studies have revealed that self-attention, a key component of Transformers, can lead to an oversmoothing issue. We show that Transformers for tabular data also face this problem, and to address the problem, we propose a novel matrix polynomial-based self-attention layer as a substitute for the original self-attention layer, which enhances model scalability. In our experiments with three representative table learning models equipped with our proposed layer, we illustrate that the layer effectively mitigates the oversmoothing problem and enhances the representation performance of the existing methods, outperforming the state-of-the-art table representation methods.

## 1 INTRODUCTION

Out of the top 10 database management systems, 7 are relational databases, including Oracle, MySQL, and Microsoft SQL Server [1]. Likewise, structured data is one of the most common data types in the fields of data mining and machine learning. With the increasing focus on tabular data, several recent methods have demonstrated remarkable success in table representation, such as (Huang et al., 2020; Ucar et al., 2021; Somepalli et al., 2021; Majmundar et al., 2022), with many of them being Transformer-based methods.

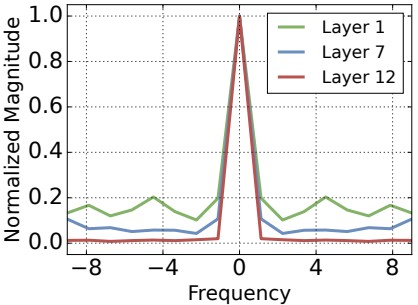

Figure 1: Spectral response of an attention map from TabTransformer (Huang et al., 2020)

Transformers have made significant advancements in deep learning, becoming state-of-the-art models in various domains, including computer vision and natural language processing (Vaswani et al., 2017; Radford et al., 2018; Devlin et al., 2019; Gulati et al., 2020; Ying et al., 2021; Dosovitskiy et al., 2021; Touvron et al., 2021; Liu et al., 2021; Rampášek et al., 2022). However, recent studies have raised concerns about the potential limitations of self-attention, a fundamental component of Transformers, specifically an issue of *oversmoothing* (Dong et al., 2021; Wang et al., 2022; Guo et al., 2023; Xue et al., 2023). Gong et al. (2021); Zhou et al. (2021) has highlighted that at deeper layers of the Transformer architecture, all token representations tend to become nearly identical (Brunner et al., 2019). The problem poses challenges when it comes to expanding the scale of training Transformers, especially in terms of depth, since Transformers rely on a simple weighted average aggregation method for value vectors.

In our preliminary experiments, we observe that Transformers designed for tabular data also exhibit the oversmoothing issue, as illustrated in Fig. 1. As we go deeper into the layers, TabTransformer (Huang et al., 2020), a model designed for tabular data, tends to focus more on low-frequency

---
[1]https://db-engines.com/en/ranking

components in its attention mechanism, even though table column relationships could be represented using a wider range of components. (For a more detailed discussion, please refer to Sec. 2.3.) To address this challenge, we propose a redesigned self-attention for table representation in this paper.

Our design is inspired by graph signal processing (GSP). In a general sense, a graph filter on a graph $\mathcal{G}$ is typically expressed as a polynomial based on its adjacency or Laplacian matrix. In the context of our work, the conventional self-attention mechanism can be considered the most basic graph filter, utilizing only $\mathbf{A}$, where $\mathbf{A} \in [0,1]^{n \times n}$ represents a learned attention matrix that encodes relationships between columns of tabular data, and $n$ is the number of input tokens. In other words, the proposed mechanism generalizes the original self-attention by allowing for more flexibility and customization. Building upon this notion, we replace the self-attention layer with our proposed polynomial-based layer, designed to approximate the optimal graph filter. In this context, we introduce our novel self-attention layer for table representation learning as **Che**byshev polynomial-based self-**Att**ention (*CheAtt*).

Our proposed self-attention is composed of coefficients $\alpha_k$ for each polynomial term and $\mathbf{A}^k$, where $k$ is the order of polynomial. It is worth noting that computing $\mathbf{A}^k$ can be computationally expensive when dealing with a large number of tokens. However, in the case of tabular data, the number of tokens is typically small because each embedded vector of a table column is considered as a token. Therefore, we can design a more scalable graph filter that utilizes the nature of tabular data.

High order polynomials require multiple squares. Here, we make use of the property of PageRank. PageRank converges after a few iterations when a transition matrix satisfies three conditions: i) stochasticity, i) irreducibility, and iii) aperiodicity. Surprisingly, attention matrices satisfy all three conditions, as discussed in Sec. 4.1. This means high order polynomial terms also converge, and thus we do not need to compute higher order polynomial terms.

Furthermore, our proposed graph filter is able to capture a wider range of frequency information as discussed in Sec. 5.2. To summarize, graph filter approximated by CheAtt encompasses both low and high-frequency components, while others often lack high-frequency signals. In summary, our contributions are as follows:

1. To the best of our knowledge, we present the first study on self-attention in the field of tabular data.

2. We propose table representation learning based on Transformer with self-attention tailored to tabular data improves representation quality compared to existing deep learning methods.

3. We have developed a Chebyshev polynomial-based self-attention mechanism that efficiently leverages properties of PageRank and self-attention matrix, without a substantial increase in computational cost.

## 2 RELATED WORK

### 2.1 REPRESENTATION LEARNING FOR TABULAR DATA

Representation learning focuses on learning meaningful features from raw data. Recently, there has been a growing focus on representation learning for tabular data. The challenges in table representation learning stem from the absence of common correlation structure in tabular data unlike the case of image and text data (Yoon et al., 2020). VIME (Yoon et al., 2020) is an approach to self- and semi-supervised learning tailored for tabular data. It incorporates a unique pretext task focused on estimating mask vectors from corrupted tabular data, along with the reconstruction pretext task. SubTab (Ucar et al., 2021) is a self-supervised learning framework designed for tabular data, which partitions the input features into multiple subsets, enhancing its ability to capture more efficient latent representations.

Transformer-based models have emerged as dominant approaches for learning useful features for tabular data (Majmundar et al., 2022; Somepalli et al., 2021; Huang et al., 2020). Tabtransformer (Huang et al., 2020) employs a Transformer encoder to acquire contextual embeddings for only categorical features. SAINT (Somepalli et al., 2021) maps both continuous and categorical features into an embedding space and then processes them through the Transformer blocks. SAINT utilizes contrastive learning and performs attention over both rows and columns to get enhanced embeddings. MET (Majmundar et al., 2022) is a table representation model based on masked autoen-

coders. It employs encoders with random masking to acquire positional embeddings for individual feature coordinates, enabling the capture of latent structures among these coordinates. These days, representations of tables are used to improve the performance of downstream tasks for tabular data, such as classification and regression, where deep learning models are struggling to beat traditional machine learning approaches.

## 2.2 Self-Attention Mechanism in Transformers

Self-Attention mechanism is the key components of Transformer architecture. Each input embedding is projected onto three parametric matrices: key, query, and value matrices, denoted as $\mathbf{K} \in \mathbb{R}^{n \times d}$, $\mathbf{Q} \in \mathbb{R}^{n \times d}$ and $\mathbf{V} \in \mathbb{R}^{n \times d}$, respectively. The self-attention mechanism $SA$ can be expressed as follows:

$$SA(\mathbf{Q}, \mathbf{K}, \mathbf{V}) = softmax(\frac{\mathbf{Q}\mathbf{K}^T}{\sqrt{d}})\mathbf{V} = \mathbf{A}\mathbf{V}, \tag{1}$$

where $d$ is the scale factor and $n$ is the number of input tokens. The basic idea of self-attention is to establish correlations between tokens (features) by assessing similarity between their key and query representations. With the calculated attention matrix $\mathbf{A}$, which is equal to $softmax(\mathbf{Q}\mathbf{K}^T/\sqrt{d})$, a value matrix $\mathbf{V}$ is re-weighted through dot-product.

Self-Attention is known to have similar characteristics to a graph convolution network (GCN). GCNs are designed to process data that can be represented as graphs denoted as $\mathcal{G} = (\mathcal{N}, \mathcal{E})$, where $\mathcal{N}$ is a node set and $\mathcal{E}$ is a set of edges connecting node pairs. Using graph convolutional layers, they learn representations of nodes within a graph, taking into account information from their local neighborhoods. Self-attention matrix used in Transformers can be seen as a normalized adjacency matrix of tokens (Guo et al., 2023).

## 2.3 Oversmoothing in GCNs and Transformers

Oversmoothing is a phenomenon that can be observed in deep learning models, particularly in GCNs. It describes a problem where a network excessively smooths node features during the aggregation process, potentially resulting in reduced discriminative capability in node representations (Oono & Suzuki, 2020; Zhou et al., 2020; Rusch et al., 2023). In Transformer, which is similar to GCN, oversmoothing phenomenon is also observed (Wang et al., 2022; Shi et al., 2022). Unlike convolutional neural networks (CNNs), Transformers do not show performance improvements by adding more layers beyond a specific threshold. This issue arises from attention matrices that are similar to GCNs. In other words, it is a fundamental problem in Transformers, and these problems occur in Transformer-based models across different domains. Dong et al. identifies the issue of "token uniformity," which diminishes the effectiveness of Transformer-based architectures by causing all token representations to be the same (Dong et al., 2021). Shi et al. explores hierarchical fusion strategies, which adaptively combine representations from various layers to introduce diversity into the output, thereby mitigating the oversmoothing issue (Shi et al., 2022). Through experiments, we observed the same issue occurring in Transformer-based table representation models. Therefore, we aim to propose an attention matrix from the perspective of graph filters that can enhance Transformer-based table representation models.

## 3 Preliminaries

### 3.1 Graph Signal Processing

Leveraging insights from graph signal processing (GSP), we designed our new attention method, CheAtt. GSP has a close connection to discrete signal processing (DSP). In DSP, a discrete signal with a length of $n$ can be represented by a vector $\mathbf{x} \in \mathbb{R}^n$. Let $\mathbf{g} \in \mathbb{R}^n$ be a filter applying to $\mathbf{x}$. The convolution $\mathbf{x} * \mathbf{g}$ can be computed as follows:

$$\mathbf{y}_i = \sum_{j=1}^{n} \mathbf{x}_j \mathbf{g}_{i-j}, \tag{2}$$

where the index refers to the $i$-th element in each vector.

GSP can be viewed as a generalized case of DSP — in other words, DSP is a special case of GSP where a *line graph with $n$ nodes* is used and therefore, the graph Fourier transform of the line graph is identical to the discrete Fourier transform. In addition, the graph convolution filter with $n$ nodes can be written with a shift operator $\mathbf{S}$ as follows:

$$\mathbf{y} = \sum_{k=0}^{K} w_k \mathbf{S}^k \mathbf{x} = \sum_{k=0}^{K} \mathbf{V}^\intercal w_k \mathbf{\Lambda}^k \mathbf{V} \mathbf{x} = \mathbf{V}^\intercal \big( \sum_{k=0}^{K} w_k \mathbf{\Lambda}^k \big) \mathbf{V} \mathbf{x} = \mathbf{V}^\intercal g(\mathbf{\Lambda}) \mathbf{V} \mathbf{x}, \tag{3}$$

where $\mathbf{x} \in \mathbb{R}^n$ is a 1-dimensional graph signal, $K$ is the order of polynomial, and $w_k \in [-\infty, \infty]$ is a coefficient. $\mathbf{S}$ is an $n \times n$ diagonalizable[2] matrix where $(i, j)$-th element is non-zero if and only if there is an edge from node $i$ to $j$ — its diagonal elements can also be non-zeros and therefore, two representative samples of $\mathbf{S}$ are adjacency and Laplacian matrices. We note that equation 3 is a generalization of equation 2 under the context of GSP. Equation 3 can be simplified as follows:

$$\mathbf{y} = \mathbf{H}\mathbf{x}, \tag{4}$$

where the graph filter $\mathbf{H}$ is the same as $\sum_{k=0}^{K} w_k \mathbf{S}^k$ in equation 3 which is called *matrix polynomial*. We note that this graph filtering operation can be extended to $d$-dimensional cases. Therefore, the core part of the self-attention, i.e., $\mathbf{A}\mathbf{V}$, can be considered as a $d$-dimensional graph filter with $\mathbf{A}$ only, where $\mathbf{H} = \mathbf{A}$. Our goal in this paper is design an effective form of $\mathbf{H}$ considering the characteristics of tabular data.

## 3.2 PAGERANK

PageRank is an algorithm used to assess the significance of web pages by considering both the quality and quantity of links leading to them, which, in turn, influences their rankings in search engine results. We refer to the collection of web pages (or nodes) as $W$ and the network of links (or directed edges) as $E$. If a page $u$ has a link pointing to page $v$, then we say $(u, v) \in E$. We denote the number of links leading out of a page $v$ as $d_v$, and the PageRank score of page $v$ as $\pi_v$. To explain PageRank, we assume a random surfer who navigates web pages based on a transition probability matrix $M \in \mathcal{R}^{N \times N}$ and a visiting probability vector $\pi^{(t)} \in \mathcal{R}^N$, where $N$ is the total number of pages and $t$ is the current iteration. In the matrix $M$, $M_{wv}$ is equal to $1/d_v$ if page $v$ links to page $w$ and 0 otherwise. The PageRank equation can be expressed as follows:

$$\pi_v^{(t)} = (1 - \epsilon) \Big( \sum_{(w,v) \in E} \frac{\pi_w^{(t-1)}}{d_w} \Big) + \frac{\epsilon}{N}, \tag{5}$$

where $\pi_v^{(t)}$ is the iterative PageRank score of page $v$ after $t$ iterations and $\epsilon$ is reset probability, representing the probability that the random surfer randomly jumps to another page. PageRank score can be computed iteratively as shown in equation 5, and the iterative method can be viewed as the power iteration. PageRank score converges quickly when its transition matrix $M$ satisfies three conditions: i) stochasticity, ii) irreducibility, and iii) aperiodicity.

## 4 CHEBYSHEV POLYNOMIAL-BASED SELF-ATTENTION (CHEATT)

In this section, we present the details of our design in a sequential manner. We start by introducing the inspiring concept behind our design, PageRank. Following that, we delve into the matrix-polynomial for our self-attention layer. Finally, we introduce another key component of our design, Chebyshev polynomial, and discuss our design from various angles.

### 4.1 PAGERANK

PageRank scores that contain the importance of pages, converge quickly when its transition matrix satisfies three conditions, as in Theorem 1. The three conditions are as follows: i) the transition matrix must be a stochastic, ii) irreducible, and iii) aperiodic matrix. Interestingly, attention matrix $\mathbf{A}$ in Transformers meet all the 3 conditions:

---

[2]For a diagonalizable square matrix $\mathbf{S} = \mathbf{V}^\intercal \mathbf{\Lambda} \mathbf{V}$, $\mathbf{S}^k = \mathbf{V}^\intercal \mathbf{\Lambda}^k \mathbf{V}$.

Figure 2: Convergence of $\mathbf{A}^k\mathbf{V}$, where $\mathbf{A}$ is an attention matrix and $\mathbf{V}$ is a value matrix for Phishing

1. **Stochasticity**: The softmax function in Transformers ensures that attention scores are normalized, making the attention matrix stochastic because values in each column sum to 1.

2. **Irreducibility**: In Transformers, attention matrices assign a non-zero probability to focus on any part of the input sequence from any position in the output sequence — note that this is guaranteed by the softmax function (cf. equation 1). This ensures the existence of a pathway, though not always direct, connecting any position to any other, satisfying the condition of irreducibility.

3. **Aperiodicity**: The aperiodicity in Markov chains condition denotes the lack of repeating patterns. In short, the irreducible chain is aperiodic if all states have a period of 1, which means that each state has at least one self-loop. This is the case in the self-attention since the attention matrix has non-zero elements, i.e., completely connected, although some are close to zeros after the softmax function — note that a negative infinite logit is required for the softmax function to produce a zero, which is not likely in neural networks.

**Theorem 1** (Convergence of PageRank). *Define the error term as the difference between the exact PageRank score $\pi_v^*$ and the t-th PageRank score $\pi_v^{(t)}$: $Err(t) = \sum_v |\pi_v^{(t)} - \pi_v^*|$, where $\pi_v^{(t)} = (1-\epsilon)\left(\sum_{(w,v)\in E} \frac{\pi_w^{(t-1)}}{d_w}\right) + \frac{\epsilon}{N}$. Then, the total error converges within a small number of iterations. The proof is in Appendix A.*

According to Theorem 1, in other words, since the attention matrix $\mathbf{A}$ satisfies the conditions, $\mathbf{A}^k\mathbf{V}$, where $k \in \mathcal{R}$, converges with a small $k$ in matrix polynomial. More discussions are in the following section. Moreover, Fig. 2 shows the convergence of $\mathbf{A}^k\mathbf{V}$. The change of the result of $\mathbf{A}^k\mathbf{V} - \mathbf{A}^{k-1}\mathbf{V}$ quickly decreases to 0 as $k$ increases. In Appendix B, we discuss the convergence of the attention matrix in detail.

### 4.2 MATRIX POLYNOMIAL-BASED TRANSFORMER

Let $\mathbf{A} \in [0,1]^{n \times n}$, where $n$ is the number of tokens, i.e., columns, be a self-attention matrix, and $\mathbf{V}^{n \times d}$, where $d$ is dimension of each token and $\mathbf{V}$ is a value matrix. Self-attention, which can also be viewed as a simplified version of graph filters, can be extended using matrix polynomial. By extending self-attention with matrix polynomial, the extended $\mathbf{H}\mathbf{V}$ can be expressed as follows:

$$\mathbf{H}\mathbf{V} = \sum_{k=0}^{n-1} w_k \mathbf{A}^k \mathbf{V}, \tag{6}$$

where $w$ are polynomial coefficients. The extended equation requires large computation of high-order power of $\mathbf{A}$. However, due to the nature of tables, which typically have only tens of columns (tokens), the computational cost becomes manageable. The expression of matrix polynomial-based self-attention is as follows:

$$\mathbf{H}\mathbf{V} \approx w_0 \mathbf{V} + w_1 \mathbf{A}\mathbf{V} + w_2 \mathbf{A}^2\mathbf{V} + \cdots + w_j \mathbf{A}^j \mathbf{V}, \tag{7}$$

where $j$ is a point where the convergence error is tolerable with respect to an enough low bound $b$, i.e., $\|\mathbf{A}^i\mathbf{V} - \mathbf{A}^j\mathbf{V}\|_F \leq b, \forall i \geq j$. Therefore, all terms higher than $j$ are absorbed to $w_j \mathbf{A}^j\mathbf{V}$ (cf. Theorem 1 and Fig. 2). As known in the existing works, we can understand that self-attention inevitably dampens, as shown in Fig. 1, the high-frequency elements (Dong et al., 2021; Wang et al., 2022; Guo et al., 2023; Xue et al., 2023). Consequently, the original self-attention is not suitable for

tasks involving representation learning, which require capturing all forms of information from the data. Conversely, when we allow $w$ to be learned and potentially take on negative values through the model learning, the graph filter will become capable of conveying high-frequency information, as we prove with our experiments in Sec. 5.

### 4.3 CHEBYSHEV POLYNOMIAL

However, optimizing $w_k, 0 \leq k \leq j$, with equation 7 can be unstable since the set of bases, i.e., $\{\mathbf{A}^k | 0 \leq k \leq j\}$, are not orthogonal. Chebyshev polynomial can be recursively defined as $T_k(\mathbf{A}) = 2\mathbf{A}T_{k-1}(\mathbf{A}) - T_{k-2}(\mathbf{A})$ with $T_0(\mathbf{A}) = \mathbf{I}$ and $T_1(\mathbf{A}) = \mathbf{A}$. These polynomials form an orthogonal basis for $L^2([-1,1], dy/\sqrt{1-y^2})$, the Hilbert space of square integrable functions with respect to the measure $dy/\sqrt{1-y^2}$. Therefore, we use Chebyshev polynomial to stabilize the training of the coefficients. The self-attention with our expended graph filter is as follows:

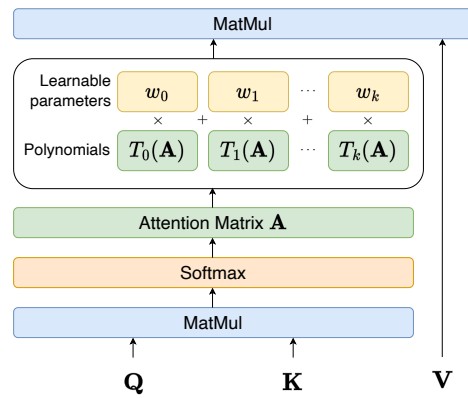

Figure 3: Architecture of the proposed CheAtt

$$\mathbf{HV} \approx \alpha_0 T_0(\mathbf{A})\mathbf{V} + \alpha_1 T_1(\mathbf{A})\mathbf{V} + \cdots + \alpha_j T_j(\mathbf{A})\mathbf{V}, \tag{8}$$

where $\alpha$ are the Chebysheb polynomial coefficients. Equation 8 can be rewritten to a polynomial of $\mathbf{A}$, since $T_k, \forall k$, is a function of $\mathbf{A}$. Thus, we utilize Chebyshev polynomial of order $j$ since the self-attention $\mathbf{A}^j\mathbf{V}$ converges rapidly with a small $j$ (cf. Theorem 1).

**Theorem 2** (Convergence of the Chebyshev coefficients Zhang & Boyd (2023); He et al. (2022b))**.** *If $f(x) = \sum_{k=0}^{\infty} \beta_k T_k(x)$, where $\beta_k$ is the Chebyshev coefficients, is weakly singular at the boundaries and analytic in the interval (-1, 1), then the Chebyshev coefficients $\beta_k$ will asymtotically (as $k \to \infty$) decreases proportionally to $1/k^q$ for some positive constant q.*

Moreover, the property of Chebyshev polynomial that Chebyshev coefficients exhibit a proportionally decreasing trend as in Theorem 2 also support a claim that we do not need to compute all $n$ terms. Theorem 2 shows Chebyshev polynomial of attention matrix does not requires computation after convergence of $\mathbf{A}^j\mathbf{V}$. In a nutshell, the proposed Chebyshev polynomial-based self-attention better approximates the graph filter without a significant increase in computation. The choice of $j$ is determined based on our preliminary experiments.

### 4.4 DISCUSSIONS

**Graph filtering aspects of CheAtt.** CheAtt enables better graph filter approximation through its Chebyshev polynomial approximation. CheAtt involves iterative matrix powers as shown in equation 8. Extensive computational resources are necessary when dealing with attention matrices in tasks that involve large datasets, such as images and graphs, which can consist of tens of thousands of tokens. For this reason, studies aiming to alleviate oversmoothing with matrix polynomial-based graph filters often limit the use of Laplacian matrix powers or optimize substitute parameter(s) to approximate the graph filter Chien et al. (2020); Gasteiger et al. (2018); He et al. (2021). In contrast, attention matrices for tables, typically containing fewer than 100 columns, involve a relatively small number of tokens, making computation more manageable. In this context, our design is more suitable for tabular data than datasets with large attention matrices.

**Apply on Transformers.** Our self-attention layer is designed by drawing inspiration from the core concepts of graph signal processing, where self-attention can be seen as a specialized form of matrix polynomial operations. It is noteworthy to emphasize that the seamless adoption of CheAtt framework into the Transformer architecture entails a simple step: the substitution of the conventional self-attention layer with our self-attention layer. This architectural substitution not only demonstrates the flexibility and compatibility of our approach but also underscores its potential to enhance the performance of existing Transformer-based models.

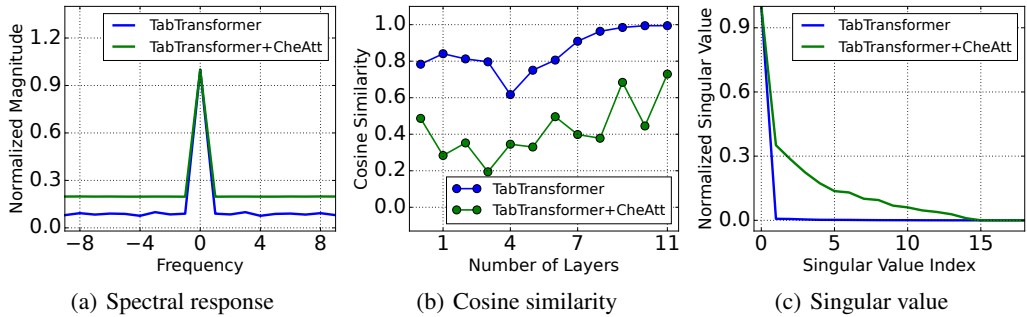

(a) Spectral response      (b) Cosine similarity      (c) Singular value

Figure 4: Visualization of spectral response, cosine similarity, and singular values of feature maps in Phishing. TabTransformer+CheAtt represents TabTransformer trained with CheAtt.

**Representation performance on the existing methods.** To validate the effectiveness of CheAtt, we apply it to the existing Transformer-based table representation models, specifically TabTransformer Huang et al. (2020), SAINT Somepalli et al. (2021), and MET Majmundar et al. (2022), with minor modifications. Details are in Appendix C. The results are in the following section.

## 5 EXPERIMENTS

### 5.1 EXPERIMENTAL ENVIRONMENTS

**Experimental settings** Our software and hardware environments are as follows: UBUNTU 20.04 LTS, PYTHON 3.8.2, PYTORCH 1.8.1, CUDA 11.4, and NVIDIA Driver 470.42.01, i9 CPU, and NVIDIA RTX A5000.

**Evaluation methods.** We use 10 datasets and 10 baselines for our experiment. Details of the datasets and baselines can be found in Appendices D.1 and D.2, respectively. To demonstrate the efficacy of CheAtt, we first compare three selected base models for table learning with base models trained using CheAtt. After training the representation models as proposed in the original paper, we subsequently train auxiliary small MLP layers for classification/regression. For classification, we report AUROC, and for regression, the reported scores are $R^2$ scores. We repeat all experiments five times and report the means and standard deviations.

### 5.2 EXPERIMENTAL RESULTS

Table 1: Comparison between base table learning models and base models trained with CheAtt. TabTransf. means TabTransformer. We report the averaged score in % across all the datasets.

|  | TabTransf. | SAINT | MET |
|---|---|---|---|
| Base model | 77.5 | 84.5 | 79.4 |
| Base model + CheAtt | **84.2** | **85.1** | **83.1** |
| Improvement | 8.65% | 0.64% | 4.66% |

Firstly, we discuss the efficacy of CheAtt. We summarize the experimental results in Table 1. As shown, CheAtt significantly improves the base models. Particularly for TabTransformer, CheAtt is highly effective, with performance increasing by an average of 8.65%. This improvement can be attributed to CheAtt's ability to capture diverse signal frequencies. In Fig. 4, (a), TabTransformer trained with CheAtt significantly retains high-frequency data compared to TabTransformer. In Fig. 4 (b), we present token-wise cosine similarity with respect to the layers. Greater cosine similarity indicates that the tokens in a layer become more similar, which is a symptom of oversmoothing. Compared to TabTransformer+CheAtt, TabTransformer exhibits higher cosine similarity in general, and as the layers get deeper, cosine similarity increases, which is also indicative of oversmoothing. In Fig. 4 (c), we present the normalized singular values of feature maps. The rapid decrease in singular values of TabTransformer indicates that the feature maps are approximately in an extremely low-

rank. On the other hand, the slow decrease in singular values of TabTransformer+CheAtt indicates that the feature maps are more representative.

## 5.3 Sensitivity on the Order of Polynomial

We perform a sensitivity experiment with respect to the order of Chebyshev polynomial, and the results are summarized in Table 2. We set the order of polynomial $k$ to 2, 3, 5, and 10. In general, for all datasets and models, we get the best scores within 5 order of polynomials. Surprisingly, MET+CheAtt performs the best at $k = 2$, unlike others shows the best score at $k = 5$. This implies that the convergence of $\mathbf{A}^j \mathbf{V}$ of MET+CheAtt at a low order is enough to represent the dataset well.

After a certain threshold of $k$, model performance tends to saturate for all models. This means that we do not need to use high-order polynomials to approximate the graph filter, as discussed above (cf. Section 4).

Table 2: Sensitivity experiment with respect to $k$. The reported scores are AUROC ($\uparrow$) for classification, and $R^2$ ($\uparrow$) for regression.

| Datasets | k | Tabtransf. + CheAtt | SAINT + CheAtt | MET + CheAtt |
|---|---|---|---|---|
| Alphabank | 2 | 61.3±1.35 | 61.4±0.31 | **62.2±0.52** |
| | 3 | 61.5±1.12 | 61.3±0.49 | 60.9±1.77 |
| | 5 | **61.6±1.01** | **62.0±0.16** | 61.4±0.46 |
| | 10 | 61.5±1.17 | 61.6±0.78 | 61.3±1.01 |
| Contraceptive | 2 | 75.9±0.75 | 75.2±0.67 | 76.7±0.59 |
| | 3 | 75.7±1.55 | 75.1±0.72 | 76.3±0.97 |
| | 5 | **76.5±1.36** | **77.1±0.38** | 70.9±13.90 |
| | 10 | 75.0±1.36 | 75.6±4.41 | **77.1±1.93** |
| Medicalcost | 2 | 86.2±0.32 | 86.5±0.18 | **86.9±0.18** |
| | 3 | 86.2±0.58 | 86.5±0.10 | 83.6±5.11 |
| | 5 | **86.8±0.41** | **86.9±0.05** | 86.1±0.56 |
| | 10 | 86.0±0.43 | 86.5±0.20 | 85.9±0.37 |

While TabTransformer+CheAtt and MET+CheAtt show robust performance on $k$, SAINT+CheAtt shows a significant decrease in performance with small $k$. In SAINT+CheAtt, when the order of the polynomial is not sufficient, the model's scalability decreases due to inappropriate approximation of the graph filter.

## 5.4 Exploring Different Polynomial Bases

We compare Chebyshev polynomial basis with others: Power, Legendre, and Jacobi polynomial, where the last three are orthogonal ones. We summarize the result in Table 3. The representation performance is robust for the type of polynomial, but Chebyshev polynomial is better than other ones in many cases. Interestingly, in the case of SAINT+CheAtt, we find that Legendre polynomial also performs well in some cases, Legendre polynomial marks the best in SAINT+CheAtt for Superconductivity. MET+CheAtt is highly dependent on polynomial, where the gap between Chebyshev and other polynomials is significant than others.

Table 3: Experimental result w.r.t. matrix polynomial forms. The reported scores are AUROC ($\uparrow$) for classification, and $R^2$ ($\uparrow$) for regression.

| Datasets | Polynomials | Tabtransf. + CheAtt | SAINT + CheAtt | MET + CheAtt |
|---|---|---|---|---|
| Default | Power | 78.8±0.29 | 78.1±0.36 | 73.1±5.63 |
| | Chebyshev | **78.9±0.09** | **78.4±0.31** | **77.8±0.14** |
| | Legendre | 78.8±0.19 | 78.1±0.23 | 58.1±24.32 |
| | Jacobi | 78.5±0.50 | 78.0±0.26 | 76.2±3.29 |
| Buddy | Power | 90.5±1.59 | 94.3±1.35 | 77.7±5.39 |
| | Chebyshev | **91.8±1.05** | **94.9±0.54** | **85.5±1.66** |
| | Legendre | 89.6±1.58 | 94.2±0.56 | 71.9±7.25 |
| | Jacobi | 90.0±2.68 | 94.3±0.63 | 79.4±7.54 |
| Super. | Power | 86.4±1.37 | 88.9±0.17 | 84.9±1.20 |
| | Chebyshev | **87.6±0.53** | 87.5±1.02 | **88.2±0.44** |
| | Legendre | 85.5±0.90 | **89.1±0.33** | 85.3±1.50 |
| | Jacobi | 68.3±0.60 | 88.8±0.33 | 85.3±0.50 |

## 5.5 Comparison to Other Methods

Table 4 presents the performances of various methods including machine learning models and deep learning models. In 6 out of 10 datasets, Transformer-based model with our attention CheAtt outperforms all baseline models. In the remaining 4 datasets, Transformer-based model with CheAtt is very close to the best model except for Activity. In Activity, ensemble models, XGBoost, and Random Forest perform better than other methods. However, among the remaining methods, excluding ensemble methods, our model consistently demonstrates the highest performance. In case of Default and Medicalcost, the Transformer-base models alone do not outperform the other methods. However, when CheAtt is incorporated into the base models, they outperform all other methods, which clearly demonstrates the effectiveness of CheAtt. In Phishing, Alphabank, Clave, and Buddy, the Transformer-base model exhibit high performance surpassing that of ensemble models, and the ad-

Table 4: Comparison with base models equipped with our proposed self-attention layer and other classification/regression models. Contra., Medical., and Super. represent Contraceptive, Medical-cost, and Superconductivity, respectively. We report AUROC ($\uparrow$) for classification and $R^2$ ($\uparrow$) for regression. The best results are in **boldface**, and the second-best results are underlined.

| Methods | Binary Classification | | | | Multi-class Classification | | | | Regression | |
|---|---|---|---|---|---|---|---|---|---|---|
| | Income | Default | Phishing | Alphabank | Clave | Contra. | Activity | Buddy | Medical. | Super. |
| MLP | 89.8±0.14 | 78.2±0.28 | 84.9±0.15 | 62.1±0.38 | 92.0±0.80 | 68.5±4.30 | 86.1±1.01 | 85.7±2.62 | 73.9±0.39 | 86.3±1.23 |
| Decision Tree | 89.5±0.07 | 76.2±0.00 | 83.1±0.00 | 60.4±0.06 | 84.8±0.17 | 75.8±0.00 | 88.5±0.23 | 82.1±0.04 | 86.8±0.00 | 83.6±0.10 |
| Regression | 57.3±0.00 | 65.1±0.00 | 85.2±0.00 | 61.5±0.00 | 91.0±0.00 | 73.6±0.00 | 68.8±0.00 | 50.0±0.00 | 74.7±0.00 | 72.3±0.00 |
| XGBoost | **92.1±0.07** | 77.5±0.21 | 82.3±0.43 | 60.5±0.54 | 95.9±0.09 | 75.0±0.56 | **98.1±0.06** | 93.5±0.30 | 80.9±1.26 | 89.9±0.10 |
| Random Forest | 91.2±0.02 | 78.6±0.15 | 85.0±0.15 | 59.8±0.15 | 93.3±0.17 | **77.3±0.08** | 98.0±0.04 | 88.5±1.34 | 86.6±0.09 | **91.3±0.06** |
| TabNet | 89.8±0.10 | 77.1±0.67 | 81.9±0.70 | 61.8±0.62 | 87.0±1.58 | 52.4±8.17 | 66.6±1.87 | 79.8±5.44 | -118.3±1.53 | 87.6±0.35 |
| VIME | 84.3±1.84 | 78.0±0.26 | 83.3±0.56 | 60.8±0.88 | 95.8±0.21 | 69.1±1.82 | 76.5±1.41 | 80.6±2.43 | 79.7±5.60 | 87.1±0.74 |
| TabTransformer | 88.9±0.87 | 78.2±0.07 | 84.2±0.35 | 59.5±1.16 | 92.9±0.85 | 64.1±1.58 | 78.0±2.54 | 85.9±2.11 | 60.1±0.09 | 83.2±0.90 |
| SAINT | 91.0±0.07 | 78.4±0.23 | 85.3±0.11 | 60.9±1.60 | 96.5±0.19 | 75.4±0.91 | 89.2±1.32 | 94.7±0.57 | 86.3±0.53 | 87.5±0.43 |
| MET | 87.8±2.63 | 76.9±0.67 | 84.5±0.44 | 61.8±0.18 | 92.9±0.25 | 76.5±1.55 | 59.0±4.66 | 84.6±1.71 | 84.8±0.57 | 85.7±0.57 |
| TabTrans.+CheAtt | 91.1±0.10 | **78.9±0.09** | **85.7±0.31** | 61.6±1.01 | 92.9±2.88 | 76.5±1.36 | 89.0±0.70 | 91.8±1.05 | 86.8±0.41 | 87.6±0.53 |
| SAINT+CheAtt | 91.3±0.05 | 78.4±0.31 | 85.6±0.46 | 62.0±0.16 | **96.5±0.11** | 77.1±0.38 | 90.5±0.40 | **94.9±0.54** | 86.9±0.05 | 87.5±1.02 |
| MET+CheAtt | 89.7±0.26 | 77.8±0.14 | 85.4±0.13 | **62.2±0.52** | 92.9±0.07 | 77.1±1.93 | 85.7±1.82 | 85.5±1.66 | **86.9±0.18** | 88.2±0.44 |

dition of CheAtt to the base model further improve its performance. This indicates that enhancing the performance of the base model can lead to the creation of even better-performing models.

## 5.6 TIME COMPLEXITY AND EMPIRICAL RUNTIME ANALYSIS

Table 5: Wall clock training time per epoch in seconds ($\downarrow$) and wall clock time for generating output representation in milliseconds ($\downarrow$)

| | Training time (per epoch) | Inference time (for 1,000 samples) |
|---|---|---|
| TabTransformer | 3.07s | 8.04ms |
| TabTrans.+CheAtt | 3.57s ($\uparrow$ 20.32%) | 9.74ms ($\uparrow$ 21.55%) |
| SAINT | 4.34s | 2.64ms |
| SAINT+CheAtt | 5.27s ($\uparrow$ 18.91%) | 3.29ms ($\uparrow$ 25.62%) |
| MET | 2.68s | 2.70ms |
| MET+CheAtt | 3.34s ($\uparrow$ 23.56%) | 3.40ms ($\uparrow$ 27.23%) |

**Time Complexity.** The original attention mechanism has a time complexity of $\mathcal{O}(n^2 d)$, where $n$ is the number of tokens and $d$ is a dimension of each token. CheAtt adds complexity to compute $A^k$ with $k-1$ matrix multiplications, resulting in a time complexity of $\mathcal{O}(n^2 d + (k-1)n^{2.371552})$, where we assume that we use algorithm in (Williams et al., 2024). Practically, if $d > (k-1)n^{2.371552}$, the time complexity of CheAtt becomes $\mathcal{O}(n^2 d)$, a condition met in almost all cases in our experiments.

**Empirical Runtime Analysis.** In Table 5, we provide a summary of the wall clock time for training and for generating output representations from dataset. For both, we report the average over all datasets. Full results are in Appendix E. For a fair comparison, we measure the time while keeping the architecture of the base models and +CheAtt models constant, and $k$ is set to 5 for CheAtt. The averaged increase across datasets in training time shows up to 24% after adapting CheAtt. For inference time, CheAtt only results in a slight increase, on the order of a few milliseconds.

## 6 CONCLUSION

Modern Transformers have revealed limitations related to oversmoothing, a phenomenon where as the depth of the Transformer model increases, hidden representations become similar for all tokens. For tabular data, we show that this problem also occurs. In order to address this phenomenon, we propose the use of Chebyshev polynomial-based self-attention, drawing inspiration from graph signal processing techniques. In our experiments, which encompassed 10 datasets and 10 baseline models, Transformer-based table representation learning models, when trained with our proposed self-attention mechanism, demonstrated significant performance improvements in downstream tasks such as classification and regression. These improvements are substantial. We anticipate that our proposed method, CheAtt, can enhance existing Transformers for table representation, paving the way for further research to delve deeper into Transformers for tabular data.

**Reproducibility Statement** To reproduce the experimental results, we have made the following efforts: 1) Source codes used in the experiments are available in the supplementary material. By following the README guidance, the main results are easily reproducible. 2) All the experiments are repeated five times, and their mean and standard deviation values are reported. 3) We provide dataset and baseline details in Appendix D.

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
