# OpenReview forum: "Polynomial-based Self-Attention for Table Representation learning"
_ICLR.cc/2024/Conference — Submitted to ICLR 2024_

### Official Review · Reviewer_uHc3 · 2023-10-24

**Soundness:** 3 good
**Presentation:** 3 good
**Contribution:** 3 good
**Rating:** 6
**Confidence:** 3

**Summary:**

The paper proposes a polynomial-based self-attention layer that enhances the representation performance of existing methods for tabular data. The experiments are thorough and convincing, showing that the proposed layer outperforms state-of-the-art methods. However, the paper lacks a detailed analysis of the computational complexity of the proposed layer and a thorough comparison with other recent approaches. Additionally, it is recommended to improve the presentation of results by adding arrows to the indicators in the charts and to test the scalability of the layer on larger datasets or more complex models.

**Strengths:**

1.The paper is well-written and the experiments are thorough and convincing.
2.The paper proposes a novel self-attention layer that enhances the representation performance of existing methods for tabular data. The experiments show that the proposed layer outperforms state-of-the-art methods.

**Weaknesses:**

Lack of detailed analysis of computational complexity. Inadequate comparison with other recent approaches. Presentation of results could be improved. Unclear scalability to larger datasets or more complex models.

**Questions:**

To further improve the quality of the manuscript, here are several suggestions:

1. The paper does not provide a detailed analysis of the computational complexity of the proposed matrix polynomial-based self-attention layer.
2. The paper does not provide a thorough comparison of the proposed approach to other methods for addressing the over-smoothing issue in Transformer-based methods for tabular data. While the experiments show that the proposed layer outperforms state-of-the-art methods, it is unclear how the proposed approach compares to other recent approaches in the literature.
3. It is recommended to add up or down arrows to the indicators in the chart, such as such as Table 2, Table 3, Table 4, and Table 5.
4. The experiments show that the layer is effective. While data set used in the experiment is small, it is unclear how the layer would scale to larger datasets or more complex models. It is recommended to increase the results of testing on large data sets.

**Details Of Ethics Concerns:**

No.

---

> ### Author Response · Authors · 2023-11-16
>
> Thank you for your thoughtful feedback and questions on our work. We have revised our manuscript following your comments.
>
> **Q1. Detailed analysis of the computational complexity**
>
> Starting with the self-attention mechanism’s time complexity, the original attention mechanism operates at $\mathcal{O}(n^2d)$, where $n$ represents the number of tokens and $d$ is the dimension of each token. Our method introduces additional complexity to compute $A^k$ with $k-1$ matrix multiplications, resulting in a time complexity of $\mathcal{O}(n^2d + (k-1)n^{2.371552})$, where we assume that we use algorithm in [1]. Practically, if $d > (k-1)n^{2.371552}$, the time complexity of CheAtt becomes $\mathcal{O}(n^2d)$, a condition observed to be met across almost all 10 datasets.
>
>
> Also, we have updated our manuscript based on your suggestion in red.
>
> **Q2. Thorough comparison of CheAtt to other methods**
>
> To the best of our knowledge, this study is the first to investigate over-smoothing in transformers applied to tabular data. Unfortunately, this uniqueness made it impossible to compare our work with other methods addressing over-smoothing in Transformer-based tabular data approaches.
>
> **Q3. Add up or down arrows in the chart**
>
> Thank you for the suggestion; we have added arrows to the tables.
>
> **Q4. How the layer would scale to larger datasets or more complex models**
>
> We emphasize that our proposed layer is applicable to any transformer-based model, regardless of its complexity. Scaling up the layer for larger or more complex datasets can be achieved through conventional methods, such as increasing the number of transformer blocks or augmenting the hidden vector size. This scaling process enables the attention matrix to expand, and the Chebyshev polynomial of the attention matrix can express a more complex data distribution. Additionally, we are conducting further experiments on larger datasets and will promptly report the results upon completion.
>
> **References**
>
> [1] Williams et al. New bounds for matrix multiplication: from alpha to omega. In SODA, 2024.

---

> > ### Comment · Reviewer_uHc3 · 2023-11-20
> > **Response to rebuttal**
> >
> > My concerns have been well addressed. I will maintain the rating.

---

### Official Review · Reviewer_BrLK · 2023-10-25

**Soundness:** 3 good
**Presentation:** 3 good
**Contribution:** 3 good
**Rating:** 8
**Confidence:** 3

**Summary:**

This paper proposes to improve the self-attention module with the matrix polynomial fashion, in order to deal with the over-smoothing issue in Transformer. The improved Transformer shows advantages in the task of understanding tabular data. The proposed polynomial-based layer, namely CheAtt, enables Transformer performs well with good efficiency due to the less-token nature of tabular data.

**Strengths:**

The paper is well written. The motivation is clear, and the proposed solution is reasonable. The experiments validate the effectiveness. The inherit issue of computational efficiency of polynomial-based layer is avoided in the task of tabular data understanding. Anyway, as far as I know, this is compatible to the current mainstream accelerating techniques.

**Weaknesses:**

It is better to present more details about the task of tabular data understanding.

**Questions:**

As I have limited experience in dealing with the tabular data, could the authors provide if any existing method tackling the issue of over-smoothing for this task?

---

> ### Author Response · Authors · 2023-11-16
>
> Thank you for your thoughtful feedback and questions on our work. We have revised our manuscript following your comments.
>
>
> **W1. Details of the task of tabular data understanding**
>
> Thank you for the comment. In response, we have included additional details about the task in our manuscript.
>
>
> **Q1. Existing research addressing the over-smoothing issue**
>
> To the best of our knowledge, our study is the first to address the issue of over-smoothing specifically for tabular data. The mitigation of over-smoothing for GCN has been a long-standing research area [1, 2]. In specific, GREAD [2] is our special case, where Equation (7) in our main paper with $k=2$, $w_0=1$, and $w_1=-1$. Moreover, research to alleviate over-smoothing and feature collapse in ViT (Vision Transformer) has recently become active [3].
>
>
> **References**
>
> [1] Rusch et al. A survey on oversmoothing in graph neural networks. arXiv preprint arXiv:2303.10993, 2023.
>
> [2] Choi et al. GREAD: Graph neural reaction-diffusion networks. In ICML, 2023.
>
> [3] Gong et al. Vision transformers with patch diversification. arXiv preprint arXiv:2104.12753, 2021.

---

> > ### Comment · Reviewer_BrLK · 2023-11-17
> > **Feedback well received.**
> >
> > My concerns are well addressed. I will remain the rating.

---

### Official Review · Reviewer_gFjL · 2023-11-01

**Soundness:** 3 good
**Presentation:** 2 fair
**Contribution:** 2 fair
**Rating:** 5
**Confidence:** 4

**Summary:**

In order to solve the over smoothing issue caused by the self-attention layer when applying a transformer to tabular data, this paper proposes Chebyshev polynomial-based self-attention. Firstly, inspired by graph signal processing, this paper considers the self-attention mechanism as a graph filter in the form of matrix polynomials and then uses finite degree Chebyshev polynomials to approximate the graph filter based on the PageRank algorithm. Experiments show that the method can effectively improve the performance of the base model without a significant increase in computation, and effectively alleviate the oversmoothing problems.

**Strengths:**

1)New ideas: This paper introduces the study of self-attention in the field of tabular data, effectively solving the oversmoothing problem.

2)Inspired new approaches: Inspired by graph signal processing and the PageRank algorithm, this paper utilizes matrix polynomials for optimizing the self-attention mechanism and uses Chebyshev polynomials to stabilize the training of coefficients.

3)Better experimental results: Experiments show that the base models, when combined with the approach in this paper, exhibit significant improvements in performance in downstream tasks such as classification and regression.

**Weaknesses:**

1）The motivation for the paper is not adequately supported by theory. The paper mentions that better flexibility and customization can be achieved by considering the self-attention mechanism as a graph filter in graph signal processing, but does not cite enough papers or theorems to fully convince this point. In addition, when proving that the self-attention matrix conforms to the three properties of the transition matrix required by the convergence of pagerank algorithm, the authors only make a rough qualitative analysis but do not carry out a more sufficient and detailed analysis and relevant theoretical or experimental proof. Since the self-attention matrix in the transformer is unpredictable, once it does not meet the corresponding requirements, the self-attention based on matrix polynomials will not be able to approximate the graph filter, and thus will not be able to realize the expected results. Therefore, I suggest the authors to provide more details in this regard.


2）Some experimental data with excessive errors will interfere with the experimental results. In the experimental part, the error range of individual experimental results is too large relative to other data. For example, 70.9±13.90 in Table 2 and 58.1±24.32 in Table 3. When these margins of error are taken into account, it becomes a question which method yields the best experimental results. This may potentially interfere with the fairness of comparisons between different methods, thereby affecting the correctness of experimental results.


3）The applicability of CheAtt should be further discussed. The paper mentioned that the effect of CheAtt is very dependent on the quality of the base model. As can be seen from Table 1 and Table 4, in TabTransformer and MET, the effect of CheAtt is outstanding, but there is almost no improvement in SAINT. They are all table representation methods based on transformer, and the original performance of SAINT is the best among the three. So what exactly does the "quality of the base model" mentioned in the paper refer to? According to the author's analysis, self-attention based on Chebyshev polynomials can improve the flexibility of the self-attention mechanism. This improvement should not be strongly related to the base model, so do the experimental results mean that CheAtt is not applicable in many situations? I suggest that the authors conduct further analysis in this area.


4）The complexity of CheAtt still needs further discussion. First, the data in Table 5 are all in the range of a few milliseconds, does it refer to the time to generate output after the model training is completed? If so, this does not take into account the large number of matrix multiplication operations required during model training. In addition, it is meaningless to only compare the absolute time difference, and it is more convincing to compare the relative time consumption. It can be seen that in Phishing dataset, the additional time spent can exceed up to 40% of the original, which is a huge and unacceptable increase. Another question is why in the MET+CheAtt method and Phishing dataset, the time after using CheAtt is reduced (from 2.7538 to 2.4625). Is this a clerical error or real experimental data? I recommend the authors to perform a more comprehensive analysis and more realistic experiments in terms of computational complexity.

**Questions:**

Please refer to the weakness above. I combined my questions with the weakness presentation.

---

> ### Author Response · Authors · 2023-11-16
> **Official Comment by Authors (1/2)**
>
> Thank you for your thoughtful feedback on our work. Following your comments, we have revised our manuscript.
>
> **W1. To include more citations, theorems, and a more rigorous analysis to substantiate the paper's motivation and claims**
>
> The self-attention matrix and the PageRank matrix share one key common characteristic that their adjacency matrices are fully connected with many small values. PageRank uses a matrix of $(1-\alpha) \mathbf{A} + \alpha\mathbf{\frac{1}{N}}$, where $\mathbf{A}$ is an adjacency matrix, $N$ is the number of nodes, and $\alpha$ is a damping factor, and therefore, all elements have non-zero (small) values. (Please refer to Section 3.2 PageRank and Equation (5) for details.) In the self-attention matrix, this is also the case since Transformers use the softmax function. Because of this characteristic, in addition, PageRank converges and so does our method.
>
> As you said, the self-attention matrix is unpredictable. As long as the self-attention matrix is fully connected, however, our method is correct (since we rely on the PageRank theory). It is practically unbelievable that the softmax produces zeros although some values are small. Note that in PageRank, small values are also used when $N$ is very large, i.e., a web-scale graph of billions of nodes.
>
> The first and second conditions shown in the paper are obvious. For the last condition, we refer to [1]. An irreducible chain, as defined in [1], has a period that is common to all states. The irreducible chain will be called $\textit{\textbf{aperiodic}}$ if all states have a period of 1. This implies that a Markov chain is aperiodic if there is at least one self-loop. As discussed earlier, the self-attention matrix has non-negative values for all elements, including diagonal elements, making the self-attention matrix aperiodic.
>
>
> We added a section in Appendix in red regarding this point.
>
>
> **W2. Results with excessive errors**
>
> For reproducibility, we repeated all experiments 5 times and reported the mean and standard deviation. We used the same seed numbers for all experiments for fair comparison. The two scores you mentioned have one or two outliers in the set. In Table 2, 70.9±13.90 is the mean and standard deviation of 73.58, 78.07, 79.44, 76.87, **46.30**. In Table 3, 58.1±24.32 is the mean and standard deviation of 77.94, **25.98**, 78.17, 77.56, **30.72**. Among those 5 repetitions, one or two scores are unusually low, attributed to the unstable training of AE.
>
>
> **W3. Improvement on SAINT**
>
> We are now testing more hyperparameters for SAINT+CheAtt since as can be seen in Appendix C.3 Table 8, we have searched only 3 hyperparameters recommended in its original paper even though SAINT has a number of hyperparameters to search. We will provide the results as soon as the ongoing experiments are completed. We will also update our paper one more time afterward.
>
> **Reference**
>
> [1] David A. et al. Markov chains and mixing times. Vol. 107. In American Mathematical Soc., 2017.

---

> ### Author Response · Authors · 2023-11-16
> **Official Comment by Authors (2/2)**
>
> **W4. Relative time consumption and more comprehensive analysis are needed**
>
> We have incorporated your suggestions into our manuscript, and we kindly ask you to refer to the updated version.
> To begin, we have discussed computational complexity in detail in our revised manuscript, marked in red. Originally, the attention mechanism operated at $\mathcal{O}(n^2d)$, where $n$ represents the number of tokens, and $d$ is the dimension of each token. Our method introduces additional complexity to compute $A^k$ with $k-1$ matrix multiplications, resulting in a time complexity of $\mathcal{O}(n^2d + (k-1)n^{2.371552})$, assuming the use of the algorithm in [2]. In practical terms, if $d > (k-1)n^{2.371552}$, the time complexity of CheAtt becomes $\mathcal{O}(n^2d)$, a condition observed across almost all 10 datasets.
>
>
> Secondly, we originally reported the absolute wall clock time to emphasize efficiency. However, we also acknowledge the importance of relative time. Therefore, in Table 5, we now present both absolute and relative time consumption. Additionally, we have updated the increase in inference time to the time taken to infer 1,000 samples. We have also included a summary of training time in Table 5, representing the average time for 5 epochs of training, along with reporting the relative time. For training time, we train the model for 5 epochs and average the time. The updated table is presented below.
>
>
> Lastly, we apologize for any confusion, but the reported reduction in time after using CheAtt on Phishing was a typo. This has been corrected in our updated paper. Thank you for bringing it to our attention.
>
> Below is the updated Table 5:
>
> |                  | Training time (per epoch)           | Inference time (for 1,000 samples)           |
> |------------------|--------------------------|---------------------------|
> | TabTransformer   | 3.07s                    | 5.42ms                    |
> | TabTrans.+CheAtt | 3.57s ($\uparrow$20.32%) | 6.26ms ($\uparrow$17.27%) |
> | SAINT            | 4.34s                    | 5.25ms                    |
> | SAINT+CheAtt     | 5.27s ($\uparrow$18.91%) | 6.68ms ($\uparrow$28.95%) |
> | MET              | 2.68s                    | 2.67ms                    |
> | MET+CheAtt       | 3.34s ($\uparrow$23.56%) | 3.47ms ($\uparrow$31.09%) |
>
> **Reference**
>
> [2] Williams et al. "New bounds for matrix multiplication: from alpha to omega." In SODA, 2024.

---

> ### Author Response · Authors · 2023-11-22
> **Final experimental result**
>
> Dear reviewer gFjL,
>
> Thank you for waiting for the final result. We have finished our additional experiments. To be more specific, we have searched for more hyperparameters for SAINT+CheAtt. The summarized result is as follows:
>
> |                   | TabTransf. | SAINT | MET   |
> |-------------------|------------|-------|-------|
> | Base model        | 77.5       | 84.5  | 79.4  |
> | Base model+CheAtt | 84.2       | 85.1  | 83.1  |
> | Improvement       | 8.65%      | 0.64% | 4.66% |
>
> The improvement of SAINT after applying CheAtt was 0.01% before we searched for more hyperparameters. After additional experiments, we achieved an improvement of 0.64%.
>
> Regarding the "quality of the base model," our initial intention was to emphasize the fact that the performance after applying CheAtt is proportional to the original performance of the base model, irrespective of CheAtt's performance. However, recognizing the potential for confusion, we have opted to remove this statement from the text.
>
> In addition, we have uploaded our new manuscript highlighting changed results in blue. Please refer to our revised manuscript following your comments.

---

### Author Response · Authors · 2023-11-16
**Change Notice**

We uploaded a revised paper with our current additional results, but we will update one more time after finishing all new experiments. The first revision will be marked in red and the second one in blue.

---

### Meta-Review · Area_Chair_YA2h · 2023-12-08

**Metareview:**

The paper proposes a Chebyshev polynomial-based self-attention mechanism to address the over-smoothing issue in Transformer models. This study is conducted in the context of learning representations of tabular data. Experiments show that the proposed attention mechanism slightly improves performance (according to Table 1) and effectively alleviates over-smoothing problems (Fig. 4).

Nonetheless, there are several issues with this submission.

First, the proposed mechanism, which improves the attention module of transformers, is supposed to alleviate the over-smoothing problem. However, as mentioned in the introduction and the related work, the over-smoothing problem is observed in many other domains. Hence, it is unclear why this work is focusing on Tabular data. Suppose the proposed mechanism is indeed helpful and removes the over-smoothing problem. In that case, it should be applied to where transformers shine the most and suffer from this smoothing problem.

Second, the performance of the resulting system, when compared to several other methods in Table 4, the proposed solution is barely better than a strong transformer-based solution (SAINT). For many tasks, the improvement is within the error bars. Moreover, for several datasets, the performance of CheAtt applied to the best transformer model lags behind simple systems such as random forests or XGBoost. On top of this, the stability of the proposed solution could be questioned. Indeed, the ablation study presented in Table 2 leads to statistically incorrect choices. The choice of the parameter k clearly shows differences with overlapping error bars.

Additionally, reviewers have raised several other concerns. The empirical evidence does not provide insights into the potential scalability of this attention module to larger datasets or more complex models. Also, the paper lacks a detailed comparative analysis and empirical comparison with other recent SOTA approaches.

For all the above reasons, this paper does not reach the publication bar of ICLR. I encourage the authors to revise the manuscript, improve the motivation and empirical evaluation, and submit it to another venue.

**Justification For Why Not Higher Score:**

The main reason is the significance of the experimental results. In many cases, when compared agains other architectures, the non transformer solutions work best, questioning the applicability of transformers for this type of data.

**Justification For Why Not Lower Score:**

N/A

---

### Decision · Program_Chairs · 2024-01-16

Reject